# What Is the Relationship between the Neighbourhood Built Environment and Time Spent in Outdoor Play? A Systematic Review

**DOI:** 10.3390/ijerph16203840

**Published:** 2019-10-11

**Authors:** Amalie Lambert, Janae Vlaar, Susan Herrington, Mariana Brussoni

**Affiliations:** 1Intern Architect, Vancouver, BC V6T 1Z2, Canada; amalie.lambert@alumni.ubc.ca; 2Human Early Learning Partnership, University of British Columbia, Suite 440, 2206 East Mall, Vancouver, BC V6T 1Z3, Canada; 3School of Architecture and Landscape Architecture, University of British Columbia, 379-2357 Main Mall, Vancouver, BC V6T 1Z4, Canada; susan.herrington@ubc.ca; 4Department of Pediatrics, School of Population and Public Health, University of British Columbia, British Columbia Children’s Hospital Research Institute, British Columbia Injury Research & Prevention Unit, F511-4480 Oak Street, Vancouver, BC V6H 3V4, Canada

**Keywords:** unstructured play, playability, child, adolescent, neighbourhood design

## Abstract

Outdoor play has been associated with children’s and adolescents’ healthy development and physical activity. Attributes of the neighbourhood built environment can influence play behaviours. This systematic review examined the relationship between attributes of the neighbourhood built environment and the time children and adolescents (0–18 years) spend in self-directed outdoor play. We identified and evaluated 18 relevant papers using the Mixed Methods Appraisal Tool and developed a narrative synthesis of study results. We found moderate evidence that lower traffic volumes (ages 6–11), yard access (ages 3–10), and increased neighbourhood greenness (ages 2–15) were positively associated with time spent in outdoor play, as well as limited evidence that specific traffic-calming street features such as fewer intersections, low traffic speeds, neighbourhood disorder, and low residential density were positively associated with time spent in outdoor play. To our knowledge, this is the first systematic review on this topic. The limited number of “good quality” studies identified highlights the need for additional research on the topic.

## 1. Introduction

Play is a central activity in children’s lives around the world [1]. Adults reflecting on favourite play memories often recall outdoor play, particularly in natural settings, remembering opportunities for freedom, fun, creativity, and skill- and confidence-building [2,3,4]. Recent research has identified a myriad benefits for children’s development, health and well-being. One of the most robust findings links time spent in outdoor play to physical activity, indicating that children playing outside are more physically active and less sedentary than when indoors [5,6]. A study of 10–13-year-old children and adolescents found that outdoor play contributed 36 min/day to physical activity, versus 40 min/day for organized sports, 17 min/day for active travel and 26 min/day for curriculum based physical activity [7]. Outdoor play has further been associated with motor, visual, and cognitive development, socio-emotional learning and mental health [8,9,10,11]. The benefits of outdoor play have also been recognized by pediatricians and public health professionals, who have stressed the importance of daily opportunities for outdoor play [12,13].

A pertinent model to consider is Sallis et al.’s ecological model for active living [14]. This model proposes that physical activity behaviours are intimately influenced by characteristics of the environment including neighbourhood design (e.g., traffic, pedestrian facilities, aesthetics), as well as how these aspects of the environment are perceived. This model helps to explain why environmental factors and parenting trends have been identified as inhibitors to children’s outdoor play, especially increased amounts of traffic, anxiety about child abduction, and increased time dedicated to academic work and structured activities [15,16,17]. The increasing amount of motorized vehicles in residential areas is frequently identified as a barrier to outdoor play [18,19,20] as high traffic speeds and volumes make both children and parents apprehensive about children being out alone [20,21,22,23]. Several international studies have also identified parents’ and children’s anxiety about children’s safety in public spaces. Parents fear their children will be abducted or hurt by abusive adults, or that their adolescents will be influenced by unruly peers [15,20,23,24,25]. In the case of neighbourhood violence, a review by Burdette and Whitaker [26] highlights that parents’ perceived danger has a greater impact on children’s outdoor play than actual crime statistics. An intervention in a US inner-city neighbourhood indicated that increasing children’s access to a play space with attendants contributed to a sense of safety, thereby resulting in more children using the space and engaging in physically active play [27]. Finally, children’s leisure time is increasingly structured, with a Dutch study coining the term “backseat generation” for children who spend much of their leisure time being driven to extracurricular activities [18]. The parenting practices, use of leisure time, and traffic characteristics mentioned above are all influenced by the neighbourhood environment in that street design and amenities help determine whether the neighbourhood is conducive to free play, feels safe, and, in turn, whether children utilize the neighbourhood for outdoor play [28,29,30].

Children play in diverse areas, depending on their age, gender, socioeconomic status and the characteristics of their neighbourhood [24]. We reviewed the literature that examines the relationship between play and the outdoor neighbourhood environment: the streets, yards and public spaces close to children’s homes. The neighbourhood built environment is the setting for a large proportion of children’s and adolescents’ outdoor play [31] and is theoretically available to individuals of all ages, abilities and socioeconomic backgrounds. Improving access to safe play spaces may provide important opportunities for reducing disparities in child health and development, particularly in urban environments [32]. A growing body of research is uncovering the influence of the built environment on health. Frank and colleagues [33] describe a conceptual framework linking transportation infrastructure, land use and walkability, the pedestrian environment and greenspace on health outcomes, including physical activity and social interaction. This framework does not incorporate child-specific considerations, yet increasing research with children highlights specific influences on child development [34]. Neighbourhood characteristics, both subjectively perceived by families and objectively measured by researchers, have been shown to impact children’s and adolescents’ access to outdoor play [35,36,37]. Previous systematic reviews have examined the relationship between the neighbourhood environment and physical activity occurring outdoors among youth [38,39]; however, a synthesis looking at exclusively outdoor play is lacking. The objective of this systematic review is to examine the relationship between physical characteristics of the neighbourhood built environment and the time children and adolescents spend in outdoor play.

## 2. Methods

This review is registered with the international prospective register of systematic reviews PROSPERO network (registration No. CRD42016046456).

### 2.1. Study Inclusion Criteria

We examined outdoor play from birth to 18 years of age. We reviewed all articles published up to 28 January 2019 (last search date). We included all study designs in the review, but only quantitative articles met our inclusion criteria. We included all articles translatable to English by Google Translate.

In this review, the independent variable is the “neighbourhood built environment attribute”, defined as a characteristic of the physical outdoor space near the residence of child or adolescent study participants, including yards, streets and public open space. We included front and back yards in the neighbourhood play environment because of their role as an intermediary between the street, the alley, and the home. Due to our interest in understanding the influence of neighbourhood planning and design, we focused on environmental characteristics, such as yard size and intersection density, as opposed to equipment such as slides and swings. We included pedestrian infrastructure such as benches, water fountains, and garbage cans because of their impact on street width. We included school grounds if they were examined outside of school hours, independently of programs such as after-school care. Because parent and child perceptions of the environment are important predictors of play-related behaviours such as child independent mobility and physical activity, we included both subjectively-perceived and objectively-measured attributes of the built environment.

Play is defined as “freely chosen, personally directed, intrinsically motivated behaviour that actively engages the child” [40]. We only included studies that examined “outdoor” play because of the numerous health and developmental benefits associated with outdoor play [16], as well as to investigate the relationship between play and the built environment. To ensure the inclusion of papers studying outdoor play behaviours in adolescents, we included the terms “hanging out”, “unstructured time” and “leisure” in our searches, the most frequent terms encountered in our survey of the literature. We decided that use of the term “play” was a sufficient descriptor of play behaviours in participants, even if the term is rarely defined in publications. Moreover, we did not include independent mobility as a type of play. Though studies have demonstrated a positive link between independent mobility and neighbourhood outdoor play [41,42], independent mobility behaviours can also include travel behaviours unrelated to play. Finally, we decided that the primary outcome of selected studies should be “time” spent in outdoor play. This criterion allowed us to identify environmental features with a measurable impact on play behaviour.

In summary, we included studies in the review if (a) they included children and adolescents aged 0–18, if (b) they reported a subjective or objective outdoor neighbourhood built environment characteristic as an independent variable, and if (c) this variable was linked to time spent in self-directed outdoor play.

### 2.2. Study Exclusion Criteria

Articles were excluded if they studied the link between outdoor play and non-neighbourhood environments, such as indoor environments. Outcome behaviours were not considered outdoor play if the activities described were explicitly directed by adults, including organized sports, time spent in school, childcare, or an after-school program. Studies which examined the link between the neighbourhood built environment and outdoor play without including a measure of time or duration were also excluded.

### 2.3. Search Strategy

The neighbourhood play search strategy is described in Appendix A. Because of the interdisciplinary scope of the topic, we searched six electronic databases during the review process: the Avery Index, MEDLINE (Ovid), PsycInfo (EBSCO), SPORTDiscus (EBSCO), ERIC (EBSCO), CINAHL (EBSCO). Additionally, we manually searched the Journal of Children, Youth and Environments because it is a journal that has traditionally published research relevant to this topic. All databases were searched on 28 January 2019. Finally, a manual search of the reference list of identified studies was conducted to search for potential additional studies, as suggested by the Cochrane Handbook for Systematic Reviews of Interventions [43]. All studies identified in the manual search were found to be duplicates of studies identified by the database search.

Search terms included Boolean combinations of the following words: play, leisure, hanging out, unstructured time, built environment, physical environment, neighbourhood, street, yard, child, youth, adolescent, teen, time and duration (Appendix A). Two independent reviewers examined the titles and abstracts of all articles identified by the search strategy and read all selected full text articles. Any discrepancies were resolved by discussion and consensus, including a third reviewer when necessary. Consensus was reached on all decisions of study eligibility.

### 2.4. Data Extraction and Quality Assessment

Data extraction was completed by one researcher (AL) and checked by another (JV). The strength of association between dependent and independent variables was extracted for each selected study, and methodological quality of studies was assessed by one researcher (JV). Another researcher (MB) reviewed all extracted effect magnitudes and checked a subset of the methodological quality assessment. When data appeared to be incomplete or incorrect, corresponding authors were contacted by email for additional information. Data extraction identified all study methods and results related to the inclusion criteria, and transcribed these in Table 1, Table 2, Table 3 and Table 4, using each study’s exact wording. We selected the Mixed Methods Appraisal Tool (MMAT) to evaluate the methodological quality of included studies because it allowed for a reliable and efficient analysis of quantitative-descriptive research. The tool evaluated the following criteria: participant recruitment (if the study reports how representative the sample is of the population), outcome measurements (whether outcome measures have been standardized or validated), participant controls (if demographic characteristics are reported and if the most important factors are controlled for in analyses), and response rates (>80% complete data and >60% response rate). For each criterion, the study received one star if it was reported by the study and met (yes), and zero stars if it was not reported by the study (can’t tell) or was reported, but not met (no). Studies were graded according to the number of criteria met (* = 1 criterion met, with a maximum of **** = 4 criteria met). Details on MMAT methodology are described elsewhere [44].

### 2.5. Analysis

Meta-analysis was planned for sufficiently homogeneous data with respect to statistical and methodological characteristics. Otherwise, narrative syntheses of research outcomes were conducted to highlight patterns in study methodologies and results. We chose to summarize study results as per Tompa, Trevithick and McLeod’s [58] best evidence synthesis guidelines, as per Appendix B. This method takes into account the number, the quality and the consistency of studies to evaluate the strength of the evidence on a topic [58]. According to the methodology, all medium and high quality studies are included in the narrative analysis, in order to highlight the most reliable results. Low quality studies were not included in the analysis. The results of the synthesis provide a general sense of which neighbourhood features are the most influential: often studies had different results for different age groups and genders. We used MMAT assessments for the quality component of the synthesis (one and two MMAT stars = “low quality study”, 3 stars = “medium quality study”, 4 stars = “high quality study”).

## 3. Results

From an initial result of 2876 papers, we identified 18 articles that fit our criteria, from 18 different studies, published between 2004 and 2016 (Figure 1).

### 3.1. Study Participants

As shown in Table 1, studies were conducted in the United States (6), the Netherlands (5), Australia (2), Canada (1), Germany (1), Switzerland (1), England (1) and Mexico (1), with a cumulative sample of 29,426 participants (accounting for participants included in Remmers et al.’s [53,54] cross-sectional and longitudinal research). Fourteen studies examined children (1–12 years) representing approximately 27,219 participants. Four studies examined both children and adolescents (0–17 years), with approximately 2,207 participants. Studies recruited participants through primary schools (7) [35,41,45,51,52,56,57], hospitals (3) [37,53,54], daycare centres or reschools (2) [47,55], school health providers (2) [36,55], advertisements and posters (2) [49,54], commercial address providers (2) [28,50] or a government program (1) [48]. Two studies used a combination of these methods [54,55]. One study did not refer to any participant recruitment strategy [46].

### 3.2. Study Characteristics

A variety of instruments were used to measure time spent in outdoor play (see Table 2), with parent questionnaires being the most prevalent (15) [45,47,49,50,51,52,53,54,55,57]. Other methods included parent diaries (1) [46] and child/adolescent questionnaires (2) [41,48]. The “Outdoor Playtime Checklist” [26] was employed by two studies [47,55], and ten studies used other validated methodologies [35,37,48,49,50,52,54,56,57]. Six studies used methodologies that have not yet been validated to evaluate outdoor play [28,36,41,45,46,53]. The neighbourhood built environment was evaluated by parent questionnaire (10) [28,35,36,49,50,52,53,54,56,57], research-team audits (6) [35,37,45,46,51,55], satellite image analysis (2) [36,47], database analysis (1) [37], child questionnaire (2) [41,48], census data (1) [37], with some studies using two of these methods [35,36,37]. Fifteen studies used validated methodologies to evaluate neighbourhood attributes [35,36,37,46,47,48,49,50,51,52,53,54,55,56,57]. Research designs were cross-sectional (14) [35,36,37,41,45,46,47,49,50,51,52,55,56,57], longitudinal (2) [48,53] or a combination of longitudinal and cross-sectional methods (2) [28,54].

### 3.3. Study Results

The selected articles identified several associations between the neighbourhood built environment and time spent in outdoor play, with significant results summarized in Table 3. Heterogeneity in measurement of time spent in outdoor play did not allow for meta-analysis. The results of the best evidence synthesis are summarized in Table 4. In order to synthesize findings, we grouped similar study results into eight themes and 17 attributes (subthemes) of the built environment. The themes and subthemes were developed inductively through consideration of the characteristics of the described features and labeled accordingly. For example, in subtheme “traffic volume”, we included Bringolf-Isler’s measure of parents’ perception of a “problem to play outdoors because of traffic” [36], Page’s measure of parents’ perception of “traffic safety”(a compound measure including the variable “heavy traffic”) [41], Aarts’ audits of the “presence of home zones” [45], and Lee’s measure of street segments with “low traffic volumes” [51]. This organization of results required certain theoretical assumptions, such as “home zones” being areas where traffic volumes would be low. Authors were contacted when terms were unclear. In Table 4, studies with an MMAT methodological rating below ***(“medium quality”) are in grey font. Their results are not included in the best evidence synthesis [see Appendix B for summary of guidelines].

#### 3.3.1. Public Open Space Characteristics (9 Studies)

This theme summarizes the impact of open space that is distinct from the home (as opposed to the yard). The review found moderate evidence that public open spaces had no effect on play: five medium/high-quality studies found no relationship between public open space attributes and time spent in outdoor play. However, other studies did find associations: in one medium-quality study, a higher proportion of parks, woods, and agriculture per 2.5 hectares around the home predicted more time spent in outdoor play for children aged 6 to 10 [36]. Inversely, one high-quality study associated a higher number of formal outdoor play spaces (play grounds, school yards, paved play grounds, and half pipe or skating track) with less time spent in play in the 7–12 age range [45]. The authors suggest that this result may be linked to the “street play” culture of the Netherlands, as well as to the quality of surrounding play spaces: play spaces perceived as unsafe may impede outdoor play.

#### 3.3.2. Street Characteristics (8 Studies)

This theme summarizes attributes related to street proportions and street infrastructure. The review found limited evidence that street characteristics have an influence on outdoor play: traffic calming features and pedestrian amenities appear to have a positive impact, while walkability and intersections seem to have a negative impact. Sidewalks, traffic lights, speed bumps, home zones (*woonerven* in the Netherlands: street configurations which often have speed limits of 15km/hr), roundabouts and “safe places to cross” were associated with more time spent in outdoor play in two high- and medium-quality studies [41,45]. Aarts et al. [45] found that pedestrian crossings had mixed results, while street lighting and safety islands were associated with less outdoor play. The presence of parallel parking spaces and parking lots had a positive association with outdoor play for older boys in Aarts et al. [45], who suggest that parallel parking spots can provide buffers between the street and the sidewalk/yard play area, while Lee et al. [51] found a negative association for both genders 6–11 for parking spaces, as part of a compound measure of “path obstructions”. In Lee et al. [51], pedestrian amenities such as benches and water fountains were associated with more outdoor play. Greater intersection density was associated with less time spent in outdoor play for children in one medium-quality study [51], and another high-quality study identified that the “presence of intersections” had the same association [45]. In one high-quality study, living in a “walkable neighbourhood” was linked to more outdoor play in a park, but to less outdoor play in a cul-de-sac or driveway [50]. Lee et al. [51] also found that walkability, defined as a combination of land use, street connectivity and residential density, was associated with less outdoor play.

#### 3.3.3. Traffic Characteristics (7 Studies)

The review found moderate evidence that low traffic volumes have a positive impact on outdoor play. Three high- and medium-quality studies reported that low traffic volumes were associated with more time spent in outdoor play [41,45,51], and one study found that traffic was a barrier to outdoor play [36]. Other subgroups were not affected: Bringolf-Isler et al. [36] found that traffic volume had no effect on the outdoor play of 13–14 year olds, Page et al. [41] had the same result for boys 10–11, as did Aarts et al. [45] for girls 4–6 and 7–12 year olds of both genders. Aarts et al. [45] found that home zones were linked to increased outdoor play, while the presence of 30 km/hr zones were associated with less outdoor play, and found no effect on girls aged 4–12.

#### 3.3.4. Housing Characteristics (7 Studies)

This theme summarizes the effects of living in a specific type of housing, living while surrounded by a certain type of housing, or living in a certain population density, a characteristic closely related to building type. The review found limited evidence linking housing and outdoor play: both higher density and detached homes were associated with less play, while duplexes had the opposite effect. Kimbro et al. [37] found that living in public housing was linked to more outdoor play in five year olds. Aarts et al. [35] found that girls 4–6 living in a detached residence spent less time in outdoor play, while boys 4–6 living in a rental property or a duplex spent more time in outdoor play. Kimbro et al. [37] and Aarts et al. [35] (girls 4–6, boys 10–12) found that living in an apartment was linked to less outdoor play. Greater neighbourhood population density was also linked to less time spent in outdoor play in two medium-quality studies for 6–11 year olds [36,51], though Lee et al. [51] examined residential density within a measure of neighbourhood walkability scores, wherein greater density is linked to greater walkability. Bringolf-Isler et al. [36] found that neither population or housing density had an effect on the outdoor play of 13–14 year olds.

#### 3.3.5. Yard Characteristics (6 Studies)

This theme includes all private outdoor space surrounding the home. The review found moderate evidence that yard access was positively associated with more outdoor play. One high-quality and two medium-quality studies reported that access to a yard was associated with more time spent in outdoor play [52], or that the absence of yards was linked to less time spent in outdoor play [35,36]. Aarts et al. [35] reported that the absence of a yard predicted more outdoor play in girls 4 to 6, while two studies found no relationship between yard access and outdoor play [35,36], Bringolf-Isler et al. [36] finding no effect on 13–14 year olds. One medium quality study found that yard size had no effect [49].

#### 3.3.6. Neighbourhood Greenness (4 Studies)

This theme includes measures that refer to general neighbourhood “greenness”, as opposed to a specific place such as a park. Studies identified greenness through satellite imagery [47], features such as street trees, flower beds and recreational areas [48] or governmental designation as a neighbourhood type with extensive greenery [35]. The review found moderate evidence that neighbourhood greenness was a predictor of more time spent in outdoor play, as identified in three high- and medium-quality studies [35,48,53]. One longitudinal study found that a perceived increase of “nature” in the neighbourhood was linked to less time spent in walking (for leisure) for boys 12–15 [48].

#### 3.3.7. Physical Disorder (4 Studies)

This theme includes measures of house maintenance, litter, graffiti, vandalism and dog feces. The review found mixed evidence of its effect on outdoor play. One medium-quality study linked greater physical disorder around the home with more outdoor play in five year-olds [37], while one high-quality study found that better house maintenance was associated with boys 10–12 spending less time in outdoor play [45]. Both studies suggest that increased physical disorder in a neighbourhood can support play. Two medium-quality studies found no association between disorder and time spent in outdoor play [54].

Other built environment attributes significantly associated with outdoor play in the 18 selected studies include the presence of water, access to a diversity of travel routes and living in a city centre [35]. Because these results were not mentioned in more than one selected study, they are not included here (see Table 3).

### 3.4. Summary of Results

The 18 studies that met our inclusion criteria examined a broad range of neighbourhood built environment features. In most studies, play behaviours were subjectively reported by parents, thus results depended on their individual interpretation of the term “play” (see Table 2). We did not identify any publications with objective measures of neighbourhood play. According to the MMAT methodological appraisal, six studies met the criteria for low methodological quality, eight studies met medium quality criteria and four studies met high quality criteria (see Table 3).

Overall, the review revealed that modifiable environmental neighbourhood features are associated with the time children and adolescents spend in outdoor play. According to the best evidence synthesis guidelines [58], no strong evidence was found in this review. However, we found moderate evidence that children and adolescents spent more time in outdoor play if they lived in neighbourhoods with low traffic volumes (ages 6–11), access to a yard (ages 3–10) and increased neighbourhood greenness (ages 2–15). Surprisingly, we found moderate evidence that access to public open space was not associated with time spent in outdoor play (ages 0–16). In this review, most studies described public open space as parks or playgrounds. Limited evidence linked time spent in outdoor play and: street features (ages 4–12), fewer intersections (ages 4–12), low walkability (ages 4–18), low traffic speeds (ages 5–12), living in rental housing (ages 4–6), living in public housing (age 5), living in a duplex (boys 4–6), low building/population density (ages 5–11) and physical disorder (ages 5–12).

Several studies identified gender-specific results: concerns about traffic safety seem more likely to constrain girls’ play, and pedestrian infrastructure was more often linked to increased play in girls. Inversely, certain traffic-related features, such as parking, were linked to more time spent in outdoor play for boys, aged 7 and up. This may be related to boys’ use of paved surfaces for team games or skateboarding. In this review, adolescents’ outdoor play was less likely to be associated with built environment features. Intersections, traffic volume and building/population density had no effect on participants aged 12 and up. We also identified regional trends: neighbourhood greenness, public open space, and housing density were more likely to be studied in Europe, whereas walkability was studied exclusively in North America.

## 4. Discussion

### 4.1. Play and Urban Design

At first glance, the features of a “playable” neighbourhood identified in this review describe a rural or suburban neighbourhood, with private yards, extensive greenery and limited traffic. Suburban typologies, developed in the post-WWII era, emphasize the “nuclear family” and private spaces for children’s play [74]. Recently, extensive research on low-density neighbourhoods has shown their negative impacts on adult health by reducing active travel behaviours [75] and requiring a high consumption of fossil fuels [76], highlighting that though suburban characteristics may support child and adolescent outdoor play, they are also associated with disadvantages for other age groups.

Most current trends in urban design (such as “smart growth” and new urbanism) promote dense typologies which emphasize active travel and public transit [76]. However, this review highlights evidence that density and certain accompanying features can have a negative impact on children’s outdoor play. One approach to this dilemma may be to identify denser neighbourhood typologies which include many, if not all the playable features synthesized in this review. For example, compact urban neighbourhoods with traffic calming features, such as cul-de-sacs that remain permeable to cyclists and pedestrians, 15 km/hr speed limits, and features, such as benches and trees, that encourage chance encounters and conversation. However, more research on small-scale street features is required to determine their impact. For example, garbage cans could be perceived as either a pedestrian amenity or a path obstruction, as in Lee et al. [51]. Ready access to parks may be less important to outdoor play: this review identified that proximate green spaces such as yards and general neighbourhood greenness have a greater impact on outdoor play than public open space. This review found that apartments and detached homes were negatively associated, while duplexes were positively associated with time spent in outdoor play: providing shared outdoor space, or other “doorstep” play spaces may be one strategy for capitalizing on the positive effects and mitigating the negative effects of housing density on children’s outdoor play.

### 4.2. Outdoor Play and Physical Activity

Several existing systematic reviews examine the relationship between the built environment and children’s physical activity. A 2006 review found that children’s physical activity was positively related to the proximity of recreational facilities and schools, as well as to the presence of sidewalks, controlled intersections, accessible destinations, and public transportation [77]. Physical activity was negatively associated with a high number of roads, high traffic density, high traffic speeds, area deprivation, and perceptions of local crime. The present review found similar results, though two studies in this review found that physical disorder had a positive association with play, especially for boys [37,45]. A 2015 review [39] found surprising results concerning objectively-measured built environment attributes and children and youth’s physical activity. Elements that attempt to promote play (play facilities, playgrounds, parks, beaches, sports venues, recreational facilities, gyms) were linked to less physical activity in young girls, and pedestrian infrastructure (sidewalks, walking tracks, path lighting, traffic lights, high connectivity streets, local destinations) was linked to less physical activity in children of both genders. Inversely, the combined effect of play infrastructure and walking infrastructure was positively correlated to adolescent girls’ and boys’ physical activity. We found similar results. Five high- and medium-quality studies in the present review examined links between access to public open space and time spent in outdoor play, with mixed results: four studies found no significant association between the two, and one study found that access to a formal outdoor play facility was negatively associated with outdoor play. We also found that pedestrian infrastructure was positively linked to girls’ outdoor play (ages 4–12).

### 4.3. Subjective and Objective Results

Four subcategories in the best evidence synthesis show trends related to their method of assessment of the built environment measurement. Four of the five medium/high quality studies that found no link between public open space and outdoor play measured access to public open space subjectively, i.e., by asking parents or children if parks are located near their home. This suggests that parents’ and children’s perceptions of public open space access do not influence outdoor play. More studies that objectively measure access to public open space would be an important counterpart to these findings. All three medium/high quality studies that found a link between yard access and outdoor play also used subjective methods. Different housing typologies likely have different yard types (e.g., townhouse courtyards), and parent and child perception of these yards could be important for outdoor play. Inversely, all medium/high quality studies measuring the impact of intersections and neighbourhood greenness used objective measures. This suggests that increasing neighbourhood greenness could lead directly to increased outdoor play, which may be useful for municipal planning strategies.

### 4.4. Limitations

This study is limited by its focus on the built environment: 11 studies out of 18 examined both social and environmental characteristics and found that social factors had a greater or equal effect on outdoor play [35,36,37,41,45,47,50,53,56]. These results emphasize the importance of employing an ecological framework that examines social environments, individual and family characteristics, built environments, and their interdependence when examining the modifiers of play [78]. For example, the results associated with housing type in this review may be associated with household socioeconomic status, which may independently influence outdoor play [79]. Furthermore, built environment interventions may have different effects in different social environments [14]. Increasing accessible green space may not be effective in a neighbourhood where residents do not feel safe allowing children to play outside. Likewise, the interdependence of environmental characteristics must also be considered. For example, increasing greenery without reducing traffic volumes in a neighbourhood may have little effect on outdoor play. Second, a large diversity of methods and measures may have contributed to our mixed results: each study measured neighbourhood environments differently, and many examined a variety of outdoor play behaviours. It is also likely that different countries and communities identify built environment features in disparate ways: a Dutch woonerf, a traffic-calmed residential street, is both similar to and radically different from a Canadian cul-de-sac. The third limitation of this study lies in its analysis of compound built environment measures which combine many variables, such as “attractiveness”, “walkability”, “poor quality of action-space”, “traffic safety”, “accessibility” and “pedestrian amenities”. Because these compound measures did not report the impact of individual variables, it can be misleading to analyze their association with outdoor play. For example, in Page et al.’s [41] measure of traffic safety, it is possible that only “heavy traffic” is significant, and that “perception of safe places to cross, roads, pollution” have no impact on outdoor play. The fourth limitation of this study is the low number of adolescent participants included. Despite our inclusion of adolescent-specific play terms, we found only four articles examining the built environment and adolescent play behaviours. Fifth, this review is limited by the small number of high quality studies identified. No strong evidence was identified in the evidence synthesis. Sixth, the unstructured outdoor play of adolescents (aged 13–18 years) may differ conceptually from younger children (0–12 years). Only one study that included adolescents divided the results in applicable age groups [48], thus we were unable to meaningfully examine potential differences. Finally, we acknowledge that most of the studies in this review were conducted in high income countries, and as such the results may not reflect realities in other economic or sociocultural contexts.

## 5. Conclusions

The findings of this systematic review suggest that the environments where children and adolescents live have some associations with the time they spend in self-directed outdoor play. In this review, we found moderate evidence that low traffic volumes (ages 6–11), yard access (ages 3–10), and increased neighbourhood greenness (ages 2–15) are associated with the time children spend in outdoor play. Interestingly, we found moderate evidence that public open spaces, such as parks and formal outdoor play facilities, have no association with outdoor play time. Evidence was limited about the impact of other features on outdoor play, but features such as fewer intersections, low residential density, low traffic speeds, living in rental housing, living in public housing and higher physical disorder appeared to support outdoor play. Moreover, limited evidence suggests that street features such as sidewalks, traffic lights, speed bumps, home zones and roundabouts have a significant association with girls’ play. To our knowledge, this is the first systematic review to examine the impact of the built environment on children’s outdoor play. Through a narrative synthesis, the review identifies common trends in international research: diminished “doorstep” play space, loss of vegetation and increased traffic have important impacts on children’s outdoor play in communities around the world. Future systematic reviews should consider the qualitative aspect of the relationship between the environment and children’s outdoor play, paying special consideration to the influence of child age and gender, and what characteristics are important to the children themselves. Future research should examine different settings, including rural communities and communities with low socioeconomic status. Possible urban design interventions to improve play opportunities could include having numerous accessible play spaces near the home, increased greenery, and trees on residential streets and traffic calming measures. Moreover, interest in designing healthier communities is growing, with several cities proposing major interventions to improve citizen health [80,81]. Cities appear to have enormous potential for reducing health inequalities when they prioritize the provision of safe places to live, work and play for all citizens [80]. Indeed, marginalized and low-income children often have lesser access to safe play spaces, as well as higher rates of illness and injury [32,82]. The results of this review suggest that providing amenities such as neighbourhood vegetation, numerous proximate play spaces and low-traffic zones are important tools for policy makers and designers to support children’s outdoor play, an essential component of child development and health.

## Figures and Tables

**Figure 1 ijerph-16-03840-f001:**
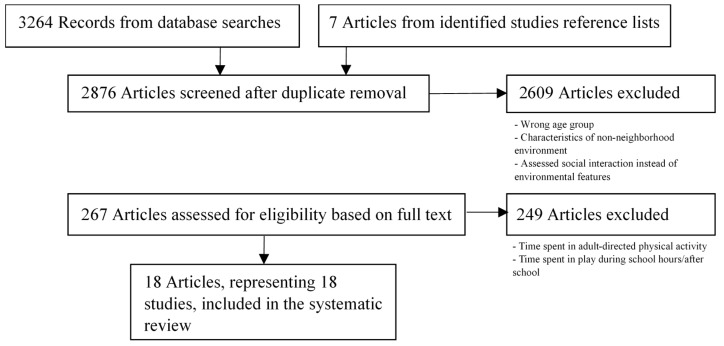
Study search strategy.

**Table 1 ijerph-16-03840-t001:** Participant characteristics.

First Author (Year)	Child Sample (*n*)	Female Children (%)	Child Age Range (Years)	Country	Location	Socio-Economic Measures	Ethnicity
Aarts (2010) [35]	6470	49.9%	4–12	Netherlands	Tilburg, Breda, ‘s-Hertogenbosch, Roosendaal	Averages of net monthly income: M (SD) 2642 (1335)–2839 (1391) EUR.	21.7–26.1% Participants with at least one biological parent not born in the Netherlands
Aarts (2012) [45]	3651	49.3%	4–12	Netherlands	Tilburg, Breda, ‘s-Hertogenbosch, Roosendaal	Parental education level: 29.6% Low 36.2% Intermediate 34.1% High	-
Blinkert (2004) [46]	430	-	5–10	Germany	Freiburg	-	-
Bringolf-Isler (2010) [36]	1081	6–10 yrs: 48.8% 13–14 yrs: 54.3%	6–14	Switzerland	Berne, Biel-Bienne, Payerne	6–10 yrs: Maternal education level: 15.8% Low 48.4% Intermediate 35.8% High 13–14 yrs: Maternal education level: 20.3% Low 48.0% Intermediate 31.8% High	6–10 yrs: 21.2% non-Swiss nationality 13–14 yrs: 22.9% non-Swiss nationality
Grigsby-Toussaint (2011) [47]	365	48%	2–5	United States	Central Illinois	Parental education level: 39% Less than a college degree	39% non-White
Gubbels (2016) [48]	208	57.6%	12–15	Netherlands	-	All participants from 20 of 140 districts considered amongst most deprived in NL	52.1% Dutch origin, 47.9% Migrant
Hales (2013) [49]	125	48.8%	3–12	United States	Chapel Hill area, University of North Carolina	Parental education level: 3.3% Did not complete high school 5.0% Completed high school 12.4% Some college 42.2% Completed a college degree 37.2% Completed a master’s degree or higher	71.3% White 25.4% African American 3.3% Other
Handy (2008) [28]	308 (cross-sectional); 272 (longitudinal)	-	0–16	United States	Northern California	54.8–78% are home owners Annual household income: 53,700–104,200 USD	-
Kercood (2015) [50]	517	49.1%	3–19	United States	Seattle, Washington and Baltimore, Maryland	Parental education level: 1.4% Did not complete high school 5.8% Completed high school 25.6% Some college or vocational training 33.5% Completed college degree 33.7% Completed graduate degree	73.3% Caucasian 14.2% African American 4.5% Asian American 1.8% Pacific Islander 0.2% American Indian 3.9% Hispanic/Mexican/Latin American 2.1% Other/not specified
Lee (2016) [51]	1321	52.6%	6–11	Mexico	Guadalajara, Mexico City, Puerto Vallarta	Annual household income: 50% < 5000 MXN 33% 5000–9999.99 MXN 17% ≥ 10,000 MXN	-
Marino (2012) [52]	2529	48.9%	3–4	United States	13.5% Northeast 23.5% Midwest 38% South 25% West	Maternal education level: 36.7% Did not complete high school 32.9% Completed high school or GED 24.1% Some college 6.3% Completed Bachelor degree or higher	22.4% White, non-Hispanic 33.1% Black, non-Hispanic 35.7% Hispanic/Latino 8.7% Other
Page (2010) [41]	1270	50.3%	10–11	England	Bristol	8/23 schools ranked in the lowest 20% of the Index of Multiple Deprivation (IMD 2000)	83.3% White 6.6% Black 4.8% Asian 4.7% mixed
Remmers (2014a) [53]	2007	49.5%	5–7	Netherlands	-	Parental education level: 2.5% Low 14.4% Mid-low 45.0% Mid-high 38.2% High	91.1% Dutch (both parents born in the Netherlands)
Remmers (2014b) [54]	1875 (cross-sectional); 1317 (longitudinal)	49.0%	5–7	Netherlands	South Netherlands, municipalities of various sizes	Maternal education level: 0.2% Low 8.7% Mid-low 38.4% Mid-high 52.7% High	-
Spurrier (2008) [55]	280	50.0%	4–6	Australia	Adelaide	8% live in low income families (annual household income < 20,000 AUD)	-
Tolbert Kimbro (2011) [37]	1822	49%	5	United States	Cohort “representative of all births in large U.S. cities in 1998–1999”	Maternal education level: 35% Did not complete high school 32% Completed high school 33% At least some college Neighbourhood poverty level: 16% High 47% Medium 37% Low	20% White 54% Black 26% Hispanic
Veitch (2010) [56]	187	47%	8–9	Australia	Melbourne	Parental education level: 18.7% < 12 yrs school 29.4% 12 yrs/trade 50.8% University	-
Veugelers (2008) [57]	5445	51.6%	10–11	Canada	Nova Scotia	Annual household income: 10.8% < 20,000 CAD 22.1% 20,000–40,000 CAD 40.5% > 60,000 CAD	-

**Table 2 ijerph-16-03840-t002:** Study characteristics.

First Author	Project	Study Design	Neighbourhood Assessment Method (Tool, e.g., ArcGIS, Monitoring, Participant Questionnaire)	Neighbourhood Size ^a^ (Neighbourhood Type) ^b^	Play Assessment Method (tool, e.g., Specific Participant Questionnaire)
Aarts [35]	-	Cross-sectional; quantitative	Parent questionnaire: perception of built environment (KOALA project, [58]) Analysis of neighbourhood type by researcher team (pre-existing databases linked to postal code)	“…area that could be reached by parents in 10–15 min by foot or in 5–8 min by bike from the respondent’s residence” (various neighbourhood types)	Parent questionnaire: “Considering a typical week in the past month: How many days does your child play outside on weekdays? On average, how long does your child spend on outdoor play on such a weekday? How many days does your child play outside on the weekend? On average, how long does your child spend on outdoor play on such a weekend day?” (“Your opinion about food and exercise”, KOALA project, [58])
Aarts [45]	-	Cross-sectional; qualitative and quantitative	Observation by trained research assistants on foot and on bicycle (Neighbourhood Environment Walkability Scale (NEWS) [59], Modified)	“Neighbourhood boundaries were defined by local databases from the municipal organization… Boundaries often coincide with physical “boundaries” such as a railway, busy road, channel or tunnel” (various neighbourhood types)	Parent questionnaire: Number of days child plays outdoors on school days and weekends. Minutes/day child involved in outdoor play on those days.
Bringolf-Isler [36]	Study on Childhood Allergy and Respiratory symptoms with respect to Air Pollution (SCARPOL)	Cross-sectional; quantitative	Parent questionnaire: perception of built environment Satellite image analysis by researcher team of street length, population and building density, green space (GIS Vector25, 2004; TwixRoute Tele Atlas, 2001)	(various neighbourhood types)	Parent questionnaire: “…parents indicated how much time their child spent on average vigorously playing outdoors on weekdays and weekends. Similar information was requested for quiet and moderately intensive play.” Comparison with accelerometer data in a subsample of 167 children (r = 0.52).
Blinkert [46]	-	Cross-sectional; quantitative	Observation by researcher team	200m radius around home (urban)	Parent diaries documenting children’s time-budgeting
Grigsby-Toussaint [47]	STRONG	Cross-sectional; quantitative	Neighbourhood greenness satellite image analysis by research team: Normalized Difference Vegetation Index (NDVI) [60]	Immediate neighbourhood is 8100 m^2^ around child’s home (various neighbourhood types)	Parent questionnaire: “…how many minutes does your child spend” on each activity of indoor active playing, indoor quiet playing, outdoor active playing, and outdoor quiet playing on an “average WEEKDAY” as well as an “average WEEKEND DAY” (Outdoor Playtime Checklist [5])
Gubbels [48]	-	Longitudinal; quantitative	Subjective assessment: NEWS questionnaire [59]; Objective assessment: District manager questionnaire on neighbourhood greenery interventions	(-)	Adolescent questionnaire: SQUASH [61,62]
Hales [49]	Home Self-administered Tool for Environmental Activity and Diet (Home-STEAD)	Cross-sectional; quantitative	Parent questionnaire on yard characteristics: natural elements checklist (15 items), presence and size of open play space, driveway, perceived sufficiency of yard space and portable equipment, and ownership of and frequency of play with dog (Home Self-administered Tool for Environmental assessment of Activity and Diet (HomeSTEAD) [49])	(-)	Parent questionnaire: asked to report number of hours the child spent playing outside. (Home Self-administered Tool for Environmental assessment of Activity and Diet (HomeSTEAD) [49])
Handy [28]	-	Quasi-longitudinal (before/after moving homes) and cross-sectional; quantitative	Parent questionnaire: perception of built environment	(various neighbourhood types)	Parent questionnaire: “If you live with children under the age of 16, how many days in the last seven days did they play outdoors somewhere in your neighbourhood (besides your backyard)?”
Kercood [50]	Neighbourhood Quality of Life Study (NQLS)	Cross-sectional; quantitative	Parent questionnaire: “How much do (neighbourhood barriers) prevent your youngest or only 4–18-year-old child from being more physically active in your neighbourhood?” (NQLS Youth Survey, [63])	(various neighbourhood types)	Parent questionnaire: “In a typical month, how often does your youngest or only 4- 18-year-old child go to the following places to play and/or be physically active?” with 12 items: *playground, playing fields/courts (e.g., baseball, basketball), skating rink, swimming pool, recreation center, a school (during non-school hours), park, vacant lot, cul-de-sac, child’s friend’s house, gym or other pay facility, skateboard park/facility*. (NQLS Youth Survey, [63])
Lee [51]	-	Cross-sectional; quantitative	Assessment by trained research team: arterial streets and residential streets (Pedestrian Environment Data Scan (PEDS) [64], Modified)	800 m around school (urban)	Parent questionnaire: number of days that a child played outdoors for ≥30 min (School PA and Nutrition (SPAN) [65], Modified)
Marino [52]	HeadStart Family and Child Experiences Survey (FACES)	Cross-sectional; quantitative	Parent questionnaire: “Is there a yard, either your own or someone else’s around your home, where [CHILD] can play?” “Is there a park or playground within walking distance of your home where [CHILD] can play?” “In the past month, has anyone in your family done the following things with [CHILD]? Visited a playground, park, or gone on a picnic?” (FACES 2006, [66])	(-)	Parent questionnaire: “We are interested in the kinds of things [CHILD] did on the last day you followed your regular routine. Did your child spend any time playing outside?” “About how much time does [CHILD] spend playing outside on a typical weekday? Would you say more than 2 h, 1–2 h, or less than 1 h?” (FACES 2006, [66])
Page [41]	Personal and Environmental Associations with Children’s Health (PEACH)	Cross-sectional; quantitative	Child questionnaire: perception of built environment	“The area where I live” (urban)	Child questionnaire: “How often do you normally play out?” Seven-point Likert scale “Playing out was described as things like riding a bike, kicking a ball around, skipping, jumping/running around, skateboarding, riding a scooter and activities that make you move around but are not structured.”
Remmers [53]	Be Active, Eat Right	Longitudinal; quantitative	Parent questionnaire: perception of physical environment (Be Active, Eat Right [67])	(-)	Parent questionnaire: asked to report number of weekdays and weekend days in an average week their child played outside. asked to indicate the average duration in the morning, noon and evening that their child played outside, again separately for weekdays and weekend days. (Be Active, Eat Right [67])
Remmers [54]	Child, parents and health: lifestyle and genetic constitution/Kind, Ouders en gezondheid: Aandacht voor Leefstijl en Aanleg (KOALA)	Longitudinal and cross-sectional; quantitative	Parent questionnaire: perception of physical aspects of the neighbourhood environment (Neighbourhood Environment Walkability Scale for Youth (NEWS-Y). [68] Modified to reflect Dutch built environment and include terms relevant to children)	(-)	Parent questionnaire (identical at child age 5 and 7 years): Play defined as: total duration of unstructured outside play in an average week, without organized sports, school physical education, and active transport. First, parents asked to report number of days their child played outside in an average week for the last four week. Second, parent asked to report the average duration of outside play.
Spurrier [55]	-	Cross-sectional; quantitative	Direct observation by researcher team (Physical and Nutritional Home Environment Inventory [55])	(-)	Parent questionnaire: “time spent by preschool children in outdoor playtime around the home and in other outdoor areas.” (Outdoor Playtime Checklist [69], Modified)
Tolbert Kimbro [37]	Fragile Families and Child Wellbeing Study (FFCWS)	Cross-sectional; quantitative	Type of housing classification by research team (U.S. Census 2000 data) Physical disorder: assessed by interviewer (Block physical disorder and physical decay measures [70], Modified)	(urban)	Parent questionnaire (Mother): “Child’s average number of hours per weekday of outdoor play” (In-Home Longitudinal Study of PreSchool Aged Children, subset of FFCWS)
Veitch [56]	-	Cross-sectional; quantitative	Parent questionnaire: perception of built environment (“Instrument to assess children’s outdoor play in various locations” [71])	(-)	Parent questionnaire: “report how often their child played in the yard at home, their own street/court/footpath, and the park/playground outside school hours on weekdays and on weekend days during a typical week.” “Responses to weekday items were marked on a five-point scale ranging from never/rarely to five days per week; and for the weekend items, on a six-point scale ranging from never/rarely to every Saturday and Sunday.” (“Instrument to assess children’s outdoor play in various locations” [71])
Veugelers [57]	Children’s Lifestyle and School-performance Study (CLASS)	Cross-sectional; quantitative	Parent questionnaire: perception of built environment (Quality of Life in Saskatoon questionnaire, [72])	School catchment areas (various neighbourhood types)	Parent questionnaire: “In the past 12 months, (outside of school hours) how often has this child: taken part in unorganized sports or physical activities without a coach or instructor?” (National Longitudinal Study of Children and Youth 2007 [73])

**Notes**: ^a^ A ‘neighbourhood size’ measure was used by several studies to define an area for analysis which reflected participants’ neighbourhood environments. ^b^ The term ‘urban’ was used by several studies to describe participants’ neighbourhood type. Other studies compared a variety of neighbourhood types. When neighbourhood type was unspecified, this is marked by (-).

**Table 3 ijerph-16-03840-t003:** Associations of built-environment measures with time spent in outdoor play.

First Author	Built Environment Attributes (Independent Variables)	Time Spent in Outdoor Play (OP) (Dependent Variable)	Measure of Association	Participant Characteristics	Strength of Association	Increase (+) or Decrease (-) in Time Spent in OP	Methodological Quality: * = 1 CRITERION Met (Low Quality) **** = 4 Criteria Met (High Quality)
Aarts [45]	Living in a semidetached or duplex (subjective: parent report) Living in a detached residence (subjective: parent report) Living in a flat or apartment (subjective: parent report) Living in a rental property (subjective: parent report) Absence of a yard (subjective: parent report) Living in a city center (objective: audit) Living in a city green area (objective: audit) Presence of water (subjective: parent report) Greater diversity of routes (subjective: parent report)	Minutes of OP per week (subjective: parent report)	Relative rate (95% CI)	Boys 4–6 Girls 4–6 Girls 4–6 Boys 10–12 Boys 4–6 Girls 4–6 Girls 7–9 Boys 7–9 Girls 4–6 Boys 4–6 Girls 7–9 Boys 10–12	RR = 1.18 (1.07, 1.29) RR = 0.86 (0.76, 0.98) RR = 0.73 (0.59, 0.89) RR = 0.77 (0.59, 0.99) RR = 1.15 (1.03, 1.28) RR = 1.13 (1.01, 1.26) RR = 0.75 (0.59, 0.95) RR = 0.79 (0.66, 0.94) RR = 1.16 (1.02, 1.31) RR = 1.04 (1.01, 1.07) RR = 1.03 (0.99, 1.06) RR = 1.08 (1.03, 1.13)	+ - - - + + - - + + + +	****
Aarts [45]	Better maintenance of houses in neighbourhood (objective: audit) Greater number of formal OP facilities per km^2^ (objective: audit) Presence of sidewalks (objective: audit) Presence of pedestrian crossings without traffic lights (objective: audit) Presence of pedestrian crossings with traffic lights (objective: audit) Presence of traffic lights (objective: audit) Presence of refuges/safety islands (objective: audit) Presence of parallel parking places (objective: audit) Presence of parking lots (grouped) (objective: audit) Presence of speed bumps (objective: audit) Presence of home zones (objective: audit) Presence of 30 km/h zones (objective: audit) Presence of roundabouts (objective: audit) Presence of intersections (objective: audit) Presence of street lighting (objective: audit)	Minutes of OP per week (subjective: parent report)	Multilevel GEE (95% CI)	Boys 10–12 Boys 7–9 Girls 7–9 Boys 10–12 Girls 10–12 Boys 4–6 Girls 4–6 Girls 10–12 Girls 4–6 Boys 7–9 Boys 4–6 Girls 7–9 Girls 7–9 Boys 7–9 Boys 10–12 Boys 10–12 Boys 7–9 Boys 7–9 Boys 4–6 Boys 7–9 Boys 10–12 Boys 4–6 Boys 7–9 Girls 7–9 Boys 10–12 Boys 4–6 Girls 4–6 Boys 7–9 Girls 7–9 Boys 10–12 Boys 4–6	RR = 0.88 (0.83–0.93) RR = 0.99 (0.99–1.00) RR = 0.99 (0.98–0.99) RR = 0.99 (0.99–1.00) RR = 0.99 (0.98–1.00) RR = 1.44 (1.16–1.18) ^c^ RR = 1.66 (1.39–1.99) RR = 1.45 (1.05–2.01) RR = 1.14 (1.01–1.28) RR = 1.20 (1.11–1.29) RR = 1.13 (1.08–1.19) RR = 0.79 (0.67–0.92) RR = 1.48 (1.28–1.72) RR = 0.89 (0.85–0.93) RR = 0.96 (0.93–1.00) RR = 1.17 (1.07–1.28) RR = 1.28 (1.18–1.38) RR = 1.25 (1.13–1.37) RR = 1.06 (1.02–1.11) RR = 0.82 (0.76–0.89) RR = 0.91 (0.86–0.97) RR = 1.14 (1.07–1.22) RR = 1.15 (1.06–1.24) RR = 1.12 (1.01–1.25) RR = 1.10 (1.04–1.16) RR = 0.82 (0.74–0.91) RR = 0.78 (0.66–0.91) RR = 0.81 (0.73–0.90) RR = 0.78 (0.69–0.88) RR = 0.87 (0.79–0.97) RR = 0.78 (0.97–0.86) ^d^	- - - - - + + + + + + - + - - + + + + - - + + + + - - - - - -	****
Blinkert [46]	Enhanced quality of action space (objective: audit) Defined as a space not possessing the following qualities of a poor action space: a speed limit of 50 km per hour, meaning normal city speed a street wider than 6 m more than four parked cars within 20 m of the door of the house no buffer separating the door from the street no public place usable by children and other pedestrians within a circle of 100 m around the home an apartment on the third floor or higher noise level higher than 50 decibels no playmates that can be reached without the help of parents no usable place for soccer, skating or other games within a circle of 200 m	Minutes per day spent outdoors without supervision (subjective: parent report)	Mean min/day	Boys and girls 5–10	“If the environment is suited for children, they spend four times longer outside without parental supervision than children who live in an environment which is not suited for them” Very poor ≈ 20 min/day Poor ≈ 35 min/day Average ≈ 60 min/day Good ≈ 75 min/day Very good ≈ 85 min/day	+	0
Bringolf-Isler [36]	Problem to play outdoors because of traffic (subjective: parent report) Problem to play outdoors because of non-availability of garden OR park (subjective: parent report) Problem to play outdoors because of non-availability of garden NOR park (subjective: parent report) Greater main street density (objective: audit) Greater population density (objective: audit) Greater building density (objective: audit) Greater presence of green spaces (objective: audit)	OP in minutes per day (subjective: parent report)	Relative difference (95% CI) Change in OP [min/day/IQR] (95% CI)	Boys and girls 6–10 Boys and girls 6–10 Boys and girls 6–10 Boys and girls 6–10 Boys and girls 6–10 Boys and girls 6–10 Boys and girls 6–10	−24.4 (−37.1, −11.6) −21.4 (−40.5, −2.4) −39.7 (−62.0, −17.5) −6.7 (−13.9, 0.5) −8.7 (−14.7, −2.7) −7.9 (−15.5, −0.2) 5.7 (−0.4, 11.9)	- - - - - - +	***
Grigsby-Toussaint [47]	Increase in neighbourhood greenness (NDVI) (objective: index)	Total outdoor PA in minutes per day (sum of child’s total “quiet” and “active” OP time) (subjective: parent report) ^a^	Linear regression (95% CI)	Boys and girls 2–5	B = 2.82 (0.21, 5.43)	+	****
Gubbels [48]	Greenery interventions (objective: manager questionnaire) Changes in perception of nature (subjective: adolescent questionnaire)	Leisure time cycling (min/week)		Boys and girls 12–15 ^e^ Boys 12–15 ^e^	B = 0.19 (*p* < 0.05) B = −0.17 (*p* < 0.05)	+ -	***
Hales [49]	Greater yard size (subjective: parent report) Greater number of natural elements in yard (subjective: parent report)	Outside play time (subjective: parent report)	Correlation coefficients	Boys and girls 3–12	No significant results		***
Handy [28]	Perception of cul-de-sac given the presence of children 6–12 (subjective: parent report) Perception of park and open space nearby (subjective: parent report) Changed (greater) perception of cul-de-sac given the presence of children 6–12 (subjective: parent report) Changed (greater) perception of large front yards given the presence of children 6–12 (subjective: parent report)	Frequency (days/week) of children’s OP in the neighbourhood (subjective: parent report)	Ordered probit regression	Boys and girls 0–16 ^e^	Coef = 0.158 (*p* = 0.015) OR = 1.17 Coef = 0.134 (*p* = 0.035) OR = 1.14 Coef = 0.170 (*p* = 0.014) OR = 1.19 Coef = 0.200 (*p* = 0.005) OR = 1.22	+ + + +	**
Kercood [50]	High walkability neighbourhood (objective: audit)	Frequency of playing or being physically active at a park (subjective: parent report) Frequency of playing or being physically active in a cul-de-sac (subjective: parent report) Frequency of playing or being physically active in their neighbours’ driveway (subjective: parent report)	Direction of association	Boys and girls 4–18 ^e^	Positive association Negative association Negative association	+ - -	****
Lee [51]	High walkability score (objective: audit) More path obstructions of street segments: poles, signs, parked cars, greenery, garbage cans (objective: audit) More street segments with low traffic volumes (objective: audit) High proportion of pedestrian amenities: public garbage cans, benches, water fountains, street vendors, vending machines (objective: audit)	Number of days of OP (≥30 min)	Logistic regression (95% CI)	Boys and girls 6–11 Boys and girls 6–11 Boys 6–11 Boys and girls 6–11	OR = 0.89 (0.82,0.98) OR = 0.43 (0.24,0.77) OR = 2.14 (1.07, 4.3) OR = 2.38 (1.24, 4.55)	- - + +	***
Marino [52]	Yard near home (subjective: parent report)	Hours per weekday of OP (subjective: parent report)	Logistic regression (95%CI)	Boys and girls 3–4	OR = 2.12 (1.41, 3.18)	+	***
Page [41]	Traffic safety: perception of safe places to cross, heavy traffic, roads, pollution (subjective: child report)	Likelihood of playing out every day (subjective: child report)	Logistic regression modelling (95% CI)	Girls 10–11	OR = 1.63 (1.14, 2.34)	+	***
Remmers [53]	Greater friendliness of physical environment for children (subjective: parent report) Greater attractiveness of physical environment for children (subjective: parent report)	Time spent in unstructured OP in minutes per week (subjective: parent report)	Linear regression		No significant results		**
Remmers [54]	Greater accessibility: “number of facilities for PA within 10 min walking distance from forest, school, playground, playing field (unpaved), gym or facility for exercise, swimming pool” (subjective: parent report)	Time spent in unstructured OP in minutes per week at age 5 and 7 years (subjective: parent report)	Repeated measures linear mixed model analyses	Boys and girls 5–7	B = 0.05 (0.01, 0.09)	+	**
Spurrier [55]	Greater backyard size (square meters) (objective: audit)	Time spent in OP (min/day) (subjective: parent report)	Pearson’s correlation	Boys and girls 4–6	r = 0.20 (*p* = 0.001)	+	**
Tolbert Kimbro [37]	Living in public housing (objective: audit) Living in an apartment (objective: audit) Greater physical disorder immediately around the home (objective: audit)	Average number of hours per weekday of OP (subjective: parent report)	Negative binomial regression	Boys and girls 5	B = 1.13 (*p* < 0.05) B = 0.88 (*p* < 0.05) B = 1.04 (*p* < 0.05)	+ - +	***
Veitch [56]	Living in cul-de-sac (subjective: parent report) Living in main arterial or busy through road (subjective: parent report)	Odds of being in the upper tertile for playing in own street/court/footpath (subjective: parent report)	Odds ratio (95% CI)	Boys and girls 8–9 Weekdays Weekends Boys and girls 8–9 Weekdays Weekends	OR = 3.99 (1.65, 9.66) OR = 3.49 (1.49, 8.16) OR = 0.84 (0.32, 2.21) OR = 0.42 (0.16, 1.12)	+ + - -	**
Veugelers [57]	Greater accessibility of parks, playgrounds, and recreational facilities (subjective: parent report)	Number of times per week spent playing sports without a coach or instructor (subjective: parent report)	Ordinal logistic regression	Boys and girls 10–11	No significant results		***

Notes: CI, 95% confidence interval; GEE, Generalized Estimating Equation; IQR, interquartile range; OP, outdoor play; OR, odds ratio; *p*, significance level; RR, relative rate/risk ratio; N.S., not significant; **^a^** The results section of Grigsby-Toussaint [47] refers to total time spent in play but the results reported in their Table 3 refers to outdoor physical activity; ^b^ Census tract data relate to poverty level, not public housing.; ^c^ This CI is likely a typographical error in the original article; ^d^ This CI is likely a typographical error in the original article; For ^c^ and ^d^ the corresponding author was contacted twice by email and no response was received; ^e^ children >12 years old.

**Table 4 ijerph-16-03840-t004:** Best evidence synthesis.

Theme	Built Environment Attribute	First author, Objectively Measured (O)/Subjectively Reported (S)	Child Age Range (Years), Gender	Methodological Quality (MMAT) ^a^	Notes	Best Evidence Synthesis ^b^
Public Open Space (POS)	Better access to POS = +OP, Less access to POS = -OP	Bringolf-I. [36] S Handy [28] S Remmers [54] S Blinkert [46] O	6–10 0–16 5–7 5–10	*** ** ** 0		Moderate evidence: 5/5 studies find no relationship between POS and OP. 1/5 studies links +POS to +OP. 1/5 studies links +POS to -OP.
	Better access to POS = -OP	Aarts [45] O	7–12	****		
	Access to POS = no effect	Aarts [45] O Bringolf-I. [36] S Marino [52] S Page [41] S Veugelers [57] S Veitch [56] S	4–6 13–14 3–4 10–11 10–11 8–9	**** *** *** *** *** **		
Street characteristics	Traffic calming street features (sidewalks, traffic lights, speed bumps, home zones, roundabouts, “safe places to cross”) = +OP	Aarts [45] O Page [41] S	4–12 10–11	**** ***	See Table 3 for details	Limited evidence: 2/2 studies link street features (sidewalks, traffic lights, speed bumps, home zones, roundabouts, “safe places to cross”) to +OP. ½ studies links safety islands and street lighting to -OP.
	Street features (safety islands, street lighting) = -OP	Aarts [45] O	4–12	****	See Table 3 for details	
	Neighbourhood walkability = +OP	Kercood [50] O	4–18	****	Play in park	Limited evidence: 2/2 studies link +walkability to -OP in driveways, cul-de-sacs and streets. 1/2 studies links +walkability to +OP in parks.
	Neighbourhood walkability = -OP	Kercood [50] O Lee [51] O	4–18 6–11	**** ***	Kercood: Play in driveway, cul-de-sac	
	Living in a cul-de-sac = +OP	Handy [28] S Veitch [56] S	0–16 8–9	** **		No evidence: no medium or high quality studies
	More intersections = -OP	Aarts [45] O Lee [51] O	4–9 10–12, M 6–11	**** ***		Limited evidence: 2/2 studies link more intersections to -OP. 1/2 studies finds no relationship between intersections and OP.
	More intersections = no effect	Aarts [45] O	10–12, F	****		
	Parking on street = +OP	Aarts [45] O	7–12, M	****		Mixed evidence: 1/2 studies links parking to +OP. 1/2 studies links parking to -OP
	Parking on street = -OP	Lee [51] O	6–11	***		
Traffic characteristics	Low traffic volumes = +OP, high traffic volumes = -OP	Aarts [45] O Bringolf-I. [36] S,O Lee [51] O Page [41] S Veitch [56] S	4–6, M 6–10 6–11 10–11, F 8–9	**** *** *** *** **		Moderate evidence: 4/4 studies link low traffic volumes to +OP. 3/4 studies find no relationship between traffic volumes and OP.
	Traffic volumes = no effect	Aarts [45] O Bringolf-I. [36] S,O Page [41] S Remmers [54] S	4–6, F 7–12 13–14 10–11, M 5–7	**** *** *** **		
	Low traffic speeds = +OP, high traffic speeds (≥30 km/hr) = −OP	Aarts [45] O Remmers [54] S Blinkert [46] O	4–12, M 5–7 5–10	**** ** 0		Limited evidence: 1/1 study links low traffic speeds to +OP. 1/1 study finds no relationship between traffic speeds and +OP.
	Low traffic speeds = no effect	Aarts [45] O	4–12, F	****		
Housing	Living in public housing = +OP	Kimbro [37] O	5	***		Limited evidence
	Living in a rental property = +OP	Aarts [35] S	4–6, M	****		Limited evidence
	Living in a semi-detached/duplex = +OP	Aarts [35] S	4–6, M	****		Limited evidence
	Living in a detached residence = −OP	Aarts [35] S	4–6, F	****		Limited evidence
	Living in an apartment = −OP	Aarts [35] S Kimbro [37] O Blinkert [46] O	4–6, F 10–12, M 5 5–10	**** *** 0		Limited evidence: 2/2 studies link living in an apartment to -OP. 1/2 studies finds no relationship between living in an apartment and OP.
	Living in an apartment = no effect	Aarts [35] S	4–6, M 7–9 10–12, F	****		
	Higher residential/building/population density = −OP	Bringolf-I. [36] O Lee [51] O	6–10 6–11	*** ***		Limited evidence: 2/2 studies link higher density to -OP. 1/2 studies finds no relationship between density and OP.
	Higher residential/building/population density = no effect	Bringolf-I. [36] O	13–14	***		
Yard access	Yard access = +OP, Absence of a yard = −OP	Aarts [35] S Bringolf-I. [36] S Marino [52] S	7–9, F 6–10 3–4	**** *** ***		Moderate evidence: 3/3 studies link yard access to +OP. 1/3 studies links yard absence to +OP. 2/3 studies find no relationship between yard access and OP.
	Absence of a yard = +OP	Aarts [35] S	4–6, F	****		
	Yard access = no effect	Aarts [35] S Bringolf-I. [36] S	4–9, M 10–12 13–14	**** ***		
Yard size	Bigger yard = +OP	Handy [28] S Spurrier [55] O	0–16 4–6	** **		Limited evidence: 1/1 study shows that yard size has no effect on OP
	Yard size = no effect	Hales [49] S	3–12	***		
Neighbourhood greenness	Greater neighbourhood greenness = +OP	Aarts [35] O Grigsby T. [47] O Gubbels O [48] Remmers [54] S	4–6, F 2–6 12–15 5–7	**** **** *** **	Cycling	Moderate evidence: 3/3 studies link neighbourhood greenness to +OP. 1/3 studies link neighbourhood greenness to -OP.
	Greater neighbourhood greenness = −OP	Gubbels [48] O	12–15, M	***	Walking	
Physical disorder	Greater physical disorder/worse neighbourhood maintenance = +OP	Aarts [45] O Kimbro [37] O	10–12, M 5	**** ***		Mixed evidence: 2/4 studies link physical disorder to +OP. 2/4 studies find no relationship between physical disorder and OP.
	Physical disorder = no effect	Aarts [45] S Page [41] S Remmers [54] S	4–9 10–12, F 10–11 5–7	**** *** **		
Noise levels	High noise levels = −OP	Blinkert [46] O	5–10	0		No evidence: no medium or high quality studies
	Noise levels = no effect	Remmers [54] O	5–7	**		

Notes: F, female; M, male; MMAT, Mixed Methods Appraisal Tool; OP, outdoor play; +OP, attributes positively associated with OP; −OP, attributes negatively associated with OP; POS, public open space; O, objectively measured built environment characteristic; S, subjectively reported built environment characteristic; ^a^ Studies with an MMAT methodological rating below *** (“medium quality”) are in grey font. Their results are not included in the best evidence synthesis; ^b^ See Appendix B for methods.

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
