# Peer review of "What Is the Relationship between the Neighbourhood Built Environment and Time Spent in Outdoor Play? A Systematic Review"

_ijerph, 2019, doi:10.3390/ijerph16203840_

Round 1

Reviewer 1 Report

I reviewed the study entitled ‘What is the relationship between the neighbourhood built environment and time spent in outdoor play? A systematic review’ with great interest. In the first round of review I may have overvalued the overestimation of study qualities as a result of applying the MMAT protocol. The high number of studies using subjective measurements as well as their focus on one-on-one associations between built environmental characteristics and outdoor play impede the strength of the outcomes. This point (particularly the one-on-one associations) could be pointed out more explicitly in the discussion section. Discussing the interplay between types of environment (e.g. social, physical) and attributes per type is warranted to prevent misinterpretation by for instance practitioners to solely focus on one specific feature within a type of environment without considering the context in which the feature would potentially be placed in. The authors addressed this issue partly in the lines 403-405, but the within-type of environment interaction has not been mentioned, which could be improved.

Starting with a clear synthesis of the evidence per theme makes the results sections much better to understand.

The inductive process towards defining 8 themes (and 17 sub-themes) can still be improved by clearly describing the process or reflect on theoretical assumptions leading to this inductive process (lines 210-211).

Author Response

We thank Reviewer 1 for their careful reading of our manuscript and thoughtful feedback. Below we outline our response to their queries. 

Reviewer 1: I reviewed the study entitled ‘What is the relationship between the neighbourhood built environment and time spent in outdoor play? A systematic review’ with great interest. In the first round of review I may have overvalued the overestimation of study qualities as a result of applying the MMAT protocol.

The high number of studies using subjective measurements as well as their focus on one-on-one associations between built environmental characteristics and outdoor play impede the strength of the outcomes. This point (particularly the one-on-one associations) could be pointed out more explicitly in the discussion section.

We have added a comment on this issue in lines 443-445: “Likewise, the interdependence of environmental characteristics must also be considered. For example, increasing greenery without reducing traffic volumes in a neighbourhood may have little effect on outdoor play.”

Discussing the interplay between types of environment (e.g. social, physical) and attributes per type is warranted to prevent misinterpretation by for instance practitioners to solely focus on one specific feature within a type of environment without considering the context in which the feature would potentially be placed in. The authors addressed this issue partly in the lines 403-405, but the within-type of environment interaction has not been mentioned, which could be improved.

We have added discussion on the interdependence of social and environmental characteristics in lines 440-443: “Furthermore, built environment interventions may have different effects in different social environments [14]. Increasing accessible green space may not be effective in a neighbourhood where residents do not feel safe allowing children to play outside.”

Starting with a clear synthesis of the evidence per theme makes the results sections much better to understand.

Thank you for your feedback.

Reviewer 2 Report

This critical review of the empirical studies on the relationship between neighbourhood built environment and children's time spent in outdoor play is nicely written.  The methodology has been clearly explained.  However, I have a few concerns and suggestions for improving the paper/study:

(1) Were the databases for literature search representative enough?  It seems to me that most of the databases used belonged to EBSCO.  Is it possible that leading journals in the field that are not indexed by EBSCO were missed?

(2) Only 18 articles were included.  Is it simply because the screening criteria are too "tight" rather than the field being ignored?

(3) Some studies (e.g. https://ajph.aphapublications.org/doi/full/10.2105/AJPH.2006.092692) that are relevant to the field were excluded.

Author Response

We thank Reviewer 2 for their careful reading of our manuscript and thoughtful feedback. Below we outline our response to their queries.

Reviewer 2: This critical review of the empirical studies on the relationship between neighbourhood built environment and children's time spent in outdoor play is nicely written.  The methodology has been clearly explained.  However, I have a few concerns and suggestions for improving the paper/study:
(1) Were the databases for literature search representative enough?  It seems to me that most of the databases used belonged to EBSCO.  Is it possible that leading journals in the field that are not indexed by EBSCO were missed?

The review aimed to examine research in the fields of architecture (Avery Index), medicine (MEDLINE), psychology (EBSCO), physiotherapy (SPORTDiscus), education (ERIC) and nursing (CINAHL). We were advised by the University’ research librarians that these databases were the correct ones to search to examine these fields. To our knowledge, there are no leading journals in these fields that are not represented in these databases.

(2) Only 18 articles were included.  Is it simply because the screening criteria are too "tight" rather than the field being ignored?

The inclusion criteria we used for this review are commensurate with the research question we sought to answer. We found that the outcome variable of time spent in outdoor play and the criterion that play had to be self-directed limited the number of relevant articles.

(3) Some studies (e.g. https://ajph.aphapublications.org/doi/full/10.2105/AJPH.2006.092692) that are relevant to the field were excluded.

Thank you for the reference. This study does not fit our inclusion criteria (the study’s outcome is not time spent in outdoor play), but is very relevant to discussions of safety and access, and has been discussed lines 65-68: “An intervention in a US inner-city neighbourhood indicated that increasing children’s access to a play space with attendants, contributed to a sense of safety, thereby resulting in more children using the space and engaging in physically active play [27].”

Reviewer 3 Report

The article concerns an interesting research problem of relationships between children outdoor play and features of the environment of their residence. I rate the article positively, but I suggest introducing a few changes. In the introduction, the authors should describe the environmental conditions of children's physical activity. A very interesting concept in this research area is the ecological model developed by Sallisa et al. In the discussion or conclusion, directions for further research in this thematic area should be formulated. An interesting problem is the potential significance of the child's age and sex, and socio-economic status family for the relationships analyzed in the article. Future research should also be extended to children living in the countryside. The authors did not formulate the strengths of the article. There is also a lack of information about the contribution of the article to world science - what is its originality in relation to previous works. I also suggest to extend practical recommendations from research. Please indicate what exactly you need to do to increase children's participation in outdoor play.

Author Response

We thank Reviewer 1 for their careful reading of our manuscript and thoughtful feedback. Below we outline our response to their queries. 

Reviewer 3: The article concerns an interesting research problem of relationships between children outdoor play and features of the environment of their residence. I rate the article positively, but I suggest introducing a few changes.

In the introduction, the authors should describe the environmental conditions of children's physical activity. A very interesting concept in this research area is the ecological model developed by Sallis et al.

Thank you for this suggestion. We have included a brief description of Sallis’ model for active living and how it gives context to the relationship between the environment and outdoor play/physical activity on lines 43-58: “A pertinent model to consider is Sallis et al.’s ecological model for active living [14]. This model proposes that physical activity behaviours are intimately influenced by characteristics of the environment including neighbourhood design (e.g., traffic, pedestrian facilities, aesthetics), as well as how these aspects of the environment are perceived. This model helps explain why environmental factors and parenting trends have been identified as inhibitors to children’s outdoor play, especially increased amounts of traffic, anxiety about child abduction, and increased time dedicated to academic work and structured activities [15–17].”  

We have also included the reference in our discussion in lines 440-443: “Furthermore, built environment interventions may have different effects in different social environments [14]. Increasing accessible green space may not be effective in a neighbourhood where residents do not feel safe allowing children to play outside.”

In the discussion or conclusion, directions for further research in this thematic area should be formulated. An interesting problem is the potential significance of the child's age and sex, and socio-economic status family for the relationships analyzed in the article. Future research should also be extended to children living in the countryside.

Directions for further research have been outlined in lines 482-486: “Future systematic reviews should consider the qualitative aspect of the relationship between the environment and children’s outdoor play, paying special consideration to the influence of child age and gender, and what characteristics are important to the children themselves. Future research, should examine different settings, including rural communities and communities with low socioeconomic status.”  

The authors did not formulate the strengths of the article. There is also a lack of information about the contribution of the article to world science - what is its originality in relation to previous works.

To our knowledge, this is the first systematic review to examine the effects of the built environment on children’s and adolescents’ play. We have added this information in the abstract, and updated our conclusion to reflect this paper’s strengths in lines 478-480: “To our knowledge, this is the first systematic review to examine the impact of the built environment on children’s outdoor play.”

I also suggest to extend practical recommendations from research. Please indicate what exactly you need to do to increase children's participation in outdoor play.

We have added recommendations in lines 479-482: “Through a narrative synthesis, the review identifies common trends in international research: diminished ‘doorstep’ play space, loss of vegetation and increased traffic have important impacts on children’s outdoor play in communities around the world.”

As well as lines 486-488: “Possible urban design interventions to improve play opportunities could include having numerous accessible play spaces near the home, increased greenery and trees on residential streets, and traffic calming measures.”

The inductive process towards defining 8 themes (and 17 sub-themes) can still be improved by clearly describing the process or reflect on theoretical assumptions leading to this inductive process (lines 210-211).

We have added further clarification of the process and our theoretical assumptions in lines 326-332: “For example, in subtheme “traffic volume”, we included Bringolf-Isler’s measure of parents’ perception of a “problem to play outdoors because of traffic” [36], Page’s measure of parents’ perception of “traffic safety”(a compound measure including the variable “heavy traffic”) [41], Aarts’ audits of the “presence of home zones” [49], and Lee’s measure of street segments with “low traffic volumes” [52]. This organization of results required certain theoretical assumptions, such as “home zones” being areas where traffic volumes would be low. Authors were contacted when terms were unclear.”

This manuscript is a resubmission of an earlier submission. The following is a list of the peer review reports and author responses from that submission.

Round 1

Reviewer 1 Report

I’ve read and reviewed the study entitled ‘What is the relationship between the neighbourhood built environment and time spent in outdoor play? A systematic review’ with great interest. The study addresses the important topic of unstructured play in the built environment. Yet, the study’s methodology contains a couple of serious flaws that impact the validity of the findings. The comments are summed below.   

Comments

Title

- Refers to systematic review, whereas narrative review seems more appropriate to the content of the manuscript.

Abstract

- 22: It remains unclear why the authors used the MMAT rather than more concise and validated tools, such as the PRISMA checklist. Further, it is unclear why the author chose to conduct a narrative review.

- 25: specific street elements is a vague description of environmental characteristics and needs further specification

Introduction

- In general well-written introduction, however recent studies on the relationship between PA and the built environment seems to lack which is likely the result of ‘updating’ the manuscript multiple times. Content-specific evidence seems to lack from 2016 on forward.

Methods

- A theoretical framework would have been helpful in more clearly specify important environmental elements that could impact PA.

- 101-104: This is a result of the search rather than a specification of the inclusion criteria. Please revise.

- 104: Study seems to lack relevant studies, also studies using objective outcome measure, such as Remmers et al., 2016 IJBNPA.

- 127-128: Describing the most recent date would be informative enough.

- 140-141: Please further specify the methodology that was applied to ensure a valid methodology 

- 140: Brackets should be the closed after (AL)

- 141: Replace ‘were’ by ‘was’

150-155: MMAT criteria need further clarification. It remains unclear how the assessment has been performed and the outcomes might be debatable. Most studies mention the low quality of studies in this field (relation BE-PA) whereas applying this tool most studies score medium or high.

- 157: Because of the disparities…. this review provides narrative syntheses of the outcomes. It is not clear why the authors chose for the narrative approach as associations described in the tables tend to overlap to a certain level based on comparable outcomes measures, narrowed to a certain time period.

-161: Only medium and high quality studies are included in the analysis. Applying this approach might lead to an overestimation of the study quality to ‘end up’ with a sufficient number of studies. To overcome this potential error, a clear description on study quality is needed.

-171-180: It is unclear what’s the value of the total number of participants. Moreover, the word ‘approximately’ seems inappropriate as numbers are very specified. The calculation of a mean sample size is irrelevant. Further, please include references in this section.

Results

- A clear distinction into age categories would enhance readability of the tables and enables to compare results for different age categories, e.g. children (<12 years old) and adolescents (12-18 years old). Their way of unstructured free play might be conceptually different as are the environmental characteristics influencing their PA in outdoor play.

- 184-193: Again, please includes references to fully understand the applied methodology per study.

- Table 1: The presented ‘location’ of the included studies is on different levels. For the Swiss, German and Dutch studies cities are mentioned, whereas the US studies refer to states (comparable to countries in Europe). Please choose either one option.

- Table 2: As outcome measure for the first study by Aarts the question ‘Your opinion about food and exercise, KOALA project’ is remarkable to choose for the assessment of outdoor play. Why is this question included.

- Table 2: In text all included studies were described as unique locations, whereas both studies of Aarts were conducted in the same environment.

- 197-202: Again, please apply an theoretical ground for the introduced subthemes. Why are these themes the most relevant?  

- 216-288: Per section a clear ‘main message’ on the association would be helpful. Or reflect on the impact of the quality of studies in relation the association between BE and PA. The description relies very much on the study of Aarts and does not reflect to broader view presented by the other included studies. Most included studies were conducted in the US, whereas the description tends to focus for a large extent to a very specific Dutch context.

290-293: The point described here reflects to the potential quality-overestimation. High-quality studies are scarce in this field, whereas this issue seems not to be valued as such in the current review. Please reflect on the as a methodological limitation.

Discussion

- 335-336: Wording issue with sentence; In this review, apartments were negatively associated… If correct, the association between both variables is negative, irrespective of the review.

- 376: Why did the authors not chose to include social environmental factors? Built environmental factors are not influencing PA behaviors in isolation, but in interplay with social environmental factors. This interplay is neglected in the current review, while on the other hand the authors propose that is important to study arrangements rather than only counting benches (413-414).

- 393-394: Due to a limited number of eligible studies it would be recommendable to delete adolescents from the scope of the review. Neither sufficient numbers of studies were found nor were associations between BE and PA clearly described. This might also be explained by the conceptually different concept outdoor PA reflects for children and adolescent, see earlier comment.  

Reviewer 2 Report

Thank you for the opportunity to review this article.  The authors provide a thorough overview of the research on this topic.

In describing the studies, it would be nice to have a category that describes locations as urban vs rural.  

I think some of the tables need to be simplified.  For example, in table 2, I don't think that "neighborhood size" adds much.  

Table 3 is overwhelming as a reader.  I think the data need to reduced for most relevant and grouped by theme as presented in table 4.  Also, sometimes N.S. data are presented (ex: Page study) and other times not (Aarts is missing many age groups).  It is not digestable in its current form.  

Line 240 references 48 twice in a row

Line 246-7 3 studies mentioned, only 2 referenced

Line 251 define home zones if not defined elsewhere

Line 272 Neighborhood greenness vs open spaces. These do not seem exclusive but associations differ.  Can you provide better definitions of these two features and how they differ or were measured?

Overall, I think the conclusions section needs work.  There are some jumps in logic that I did not see supported in the results.  New ideas are presented with no previous mention.  Examples outlined below.

Line 403 I think saying "most associated" is misleading.  

Line 410- What results would you say are the impact of urbanization?  Be more direct.

Line 413-415  Who is "we?"  If a new theory is going to be presented, it needs to be better described and introduce earlier in the paper.  How does this link back to your findings?

Line 416 - Starting with "Cities appear..." I think this idea of inequities should be introduced in the introduction if it is to be discussed.  What public amenities do you suggest based on these research?  How are these going to reduce inequities?